# Evaluation of Intranasal Midazolam for Pediatric Sedation during the Suturing of Traumatic Lacerations: A Systematic Review

**DOI:** 10.3390/children9050644

**Published:** 2022-04-29

**Authors:** Francisco Javier Gómez-Manzano, José Alberto Laredo-Aguilera, Ana Isabel Cobo-Cuenca, Joseba Rabanales-Sotos, Sergio Rodríguez-Cañamero, Noelia Martín-Espinosa, Juan Manuel Carmona-Torres

**Affiliations:** 1Facultad de Fisioterapia y Enfermería, Universidad de Castilla La Mancha, 45071 Toledo, Spain; javier.gomezmanzano@uclm.es (F.J.G.-M.); anaisabel.cobo@uclm.es (A.I.C.-C.); noelia.martin@uclm.es (N.M.-E.); juanmanuel.carmona@uclm.es (J.M.C.-T.); 2Grupo de Investigación Multidisciplinar en Cuidados, Universidad de Castilla-La Mancha, 45071 Toledo, Spain; sergio_roca5@hotmail.com; 3IES Juanelo Turriano, Consejería de Educación, Cultura y Deportes de Castilla-La Mancha, 45007 Toledo, Spain; 4Facultad de Enfermería, Universidad de Castilla-La Mancha, 02071 Albacete, Spain; joseba.rabanales@uclm.es; 5Hospital Universitario de Toledo, Servicio de Salud de Castilla-La Mancha, 45007 Toledo, Spain

**Keywords:** midazolam, intranasal administration, laceration, paediatrics, nursing

## Abstract

Objective: The objective of this study was to evaluate the efficacy and safety of intranasal midazolam as part of a paediatric sedation and analgesic procedure during the suturing of traumatic lacerations in paediatric emergency departments. Methodology: A systematic review of clinical trials was completed in July 2021. The databases consulted were PUBMED, SCOPUS, WEB OF SCIENCE, NICE and Virtual Health Library. Eligibility criteria: randomised and nonrandomised clinical trials. Two independent, blinded reviewers performed the selection and data extraction. The participants were 746 children, of whom, 377 received intranasal midazolam. All of the children were admitted to an emergency department for traumatic lacerations that required suturing. The quality of the articles was evaluated with the Jadad scale. This systematic review was conducted according to the PRISMA (Preferred Reporting Items for Systematic Reviews and Meta-Analyses) guidelines. Results: Nine studies were included in the review. The intranasal administration of midazolam in healthy children produces anxiolysis and minimal/moderate sedation without serious side effects. Although there are combinations of parenteral drugs that produce deeper sedation, they also have greater adverse effects. No significant differences in the initiation of sedation and the suture procedure were found between the intranasal route and the parenteral route. Conclusions: The use of intranasal midazolam in healthy children produces sufficiently intense and long-lasting sedation to allow for the suturing of traumatic lacerations that do not present other complications; therefore, this drug can be used effectively in paediatric emergency departments.

## 1. Introduction

In children, trauma frequently causes minor lacerations that should be evaluated at paediatric emergency departments. In the United States, lacerations account for approximately 8.2% of annual emergency department visits, which means that between seven and nine million lacerations are treated in the US each year [1].

When suturing is necessary, the fear of pain, separation from parents and refusal to be examined or remain in a certain position manifest as anxiety, crying and aggression, especially in young children. In turn, these experiences increase the suffering of the child and his or her parents, making it difficult for the health professional to perform the suture [2,3,4].

Local anaesthetics produce analgesia but lack the ability to control the child’s fear and anxiety [5,6]. However, this can be alleviated through pharmacological and nonpharmacological measures [2,6,7]. Benzodiazepines are sedative drugs that are commonly used in paediatrics for conscious sedation as part of a paediatric sedation and analgesic (PSA) procedure in various procedures [8,9]. Specifically, midazolam is a water-soluble, fast-acting, quickly eliminated GABA receptor agonist [10] that produces sedation, anxiolysis, hypnotic effects and anterograde amnesia; furthermore, it is an anticonvulsant and muscle relaxant and does not have serious side effects [5,11,12,13]. Midazolam can be administered intravenously, intramuscularly, orally, rectally or intranasally [14].

In young children, the parenteral administration of sedative drugs may not be adequate, since it increases anxiety, fear and anguish due to the puncture [5]. In addition, it requires more technical skill, and, due to the lack of cooperation of the child, it may pose a greater risk of accidental needle piercing for the health professional [15].

The intranasal route has proven to be a non-invasive, simple-to-use, safe and effective option [2]. The nasal mucosa is highly vascularised and provides direct access to the central nervous system, avoiding the first-pass effect and producing the rapid initiation of action, similar to that achieved with the intravenous route [2,6,9,16]. Currently, paediatric intranasal sedation has two clinical uses: the sedation of the child before uncomfortable or painful procedures and as premedication for general anaesthesia [11].

According to our study, although there are currently no official statistics on the number of traumatic laceration suture procedures in children, this is a common procedure in paediatric emergency departments; therefore, it is necessary for health professionals to be aware of the available alternatives and to know which anxiolytic/sedative drugs offer the best guarantees of safety and efficacy.

Therefore, the objective of this systematic review was to synthesise the available scientific evidence based on clinical trials to evaluate the efficacy and safety of intranasal midazolam as an anxiolytic and sedative as part of a paediatric sedation and analgesic (PSA) procedure during the suturing of traumatic lacerations in paediatric emergency departments.

## 2. Materials and Methods

### 2.1. Design and Sources of Information

A systematic review was conducted in accordance with the Preferred Reporting Items for Systematic Reviews and Meta-Analyses (PRISMA) statement [17]. This systematic review was registered in PROSPERO with number CR42021224635.

The databases consulted from January to July 2021 were PUBMED, SCOPUS, WEB OF SCIENCE, NICE and Virtual Health Library (VHL).

### 2.2. Search Strategy

The search strategy was guided by the PICO research questions (population, intervention, comparator, outcomes) shown in Table 1. The international databases were consulted using the following search string: (lacerations OR sutures) AND midazolam AND (nasal OR intranasal OR intra-nasal).

We consider that a drug is “effective” if it produces a rapid state of sedation in the child allowing for the successful suturing of a traumatic laceration without complications in the emergency department. Likewise, we consider that a drug is “safe” if we observe that the doses administered do not produce significant adverse effects or require a prolonged recovery of the child until they are discharged from hospital.

### 2.3. Inclusion and Exclusion Criteria

Table 2 shows the inclusion and exclusion criteria used to select the studies. Only interventional clinical trials were included in an effort to obtain the highest possible quality of scientific evidence.

### 2.4. Study Selection

Two researchers (F.J.G.-M. and J.A.L.-A.) independently and blinded to one another selected the studies according to the predetermined inclusion and exclusion criteria and evaluated all of the references of studies identified in the search. After searching the databases mentioned above, a total of 99 articles were obtained. These potentially relevant articles were imported to the Mendeley reference manager to eliminate duplicates (35 studies), in accordance with the PRISMA Systematic Review Flowchart (Figure 1). Once duplicates were eliminated, screening of the titles and abstracts was performed. The articles that met the inclusion criteria during this first phase were read in full to determine their eligibility for inclusion in the final sample. In cases of doubt or discrepancy, a third author (J.M.C.-T.) was consulted.

### 2.5. Evaluation of the Quality of the Studies

Quality was assessed using the Jadad scale [18]. This tool evaluates the quality of the studies using a score of 0 (low quality) to 5 points (excellent quality). No eligible study was excluded from the review due to its quality.

All included studies underwent quality assessment and were independently evaluated by two reviewers (F.J.G.-M. and A.I.C.-C.). The disagreements and discrepancies were discussed until consensus was reached. If consensus was not achieved, another reviewer was consulted (J.M.C.-T.). 

### 2.6. Data Extraction

For data extraction, two authors (F.J.G.-M. and J.M.C.-T.) independently used a standardised data collection form. From each selected study, the following data were collected: (1) the name of the first author and year of publication; (2) country; (3) study design and type of intervention; (4) characteristics of the sample: sample size, selection, type of blinding, age of participants and size of the laceration; (5) characteristics of the intervention: dose, drug, route of administration, intranasal method of administration and vital signs; and (6) main results of the intervention: level of sedation/anxiolysis, time until the start of sedation, time until discharge from the hospital, observed adverse effects and degree of satisfaction of parents/health professionals.

### 2.7. Analysis of the Data Obtained

A narrative synthesis of each of the studies included in this systematic review was performed. The data were analysed to compare the results for intranasal midazolam with those for different sedative drugs and placebo (Table 3) based on sample size, dose used, method of midazolam administration, level of sedation/anxiolysis, time until the initiation of sedation, time until discharge from the hospital, observed adverse effects and degree of satisfaction of parents/health professionals (Table 4).

## 3. Results

Of the 99 articles that were obtained, nine clinical trials were finally selected [5,7,8,10,16,19,20,21,22] (Figure 1). The selected articles included a total of 746 children, of whom, 377 received intranasal midazolam. Six clinical trials included children under 7 years of age, and three studies included children up to 12 years of age. All children were admitted to a paediatric emergency department for traumatic lacerations without additional complications.

The studies had a randomised clinical trial (RCT) [7,8,10,16,19,20,21,22] or unrandomised design [5]. Table 5 shows the characteristics and the most relevant data of the studies included in this review.

### 3.1. Risk of Bias

The Jadad scale was used to assess the methodological quality of the studies and detect bias [18]. According to this scale, seven of the nine included articles were of good or excellent quality. Table 4 shows the score for each clinical trial included in this review.

### 3.2. Intranasal Midazolam versus Placebo and Control 

Theroux M. et al., (1993) assessed the degree of crying, struggling and movement by children during the suturing procedure [19]. The group that received intranasal midazolam cried less (*p* < 0.003), struggled less (*p* < 0.04) and had a significantly lower maximum heart rate (*p* < 0.002) and maximum systolic blood pressure (*p* < 0.04) than the placebo/control group [20].

### 3.3. Intranasal Midazolam versus Oral and Intraoral Midazolam

No significant differences were observed in the level of anxiety reduction [8,20,21] or the total length of stay in the hospital [8,20]. The intranasal route was the least tolerated because it produced burning or nasal irritation [8,20,21].

According to Klein et al., (2011), the intranasal route presents a faster initiation of action, a higher proportion of adequately sedated children, an optimal score on the activity scale that was used and a greater proportion of parents who would choose this method of sedation in the future [21].

No serious adverse effects were observed with any of the examined routes of administration [8,20,21].

### 3.4. Intranasal Midazolam versus Intramuscular Ketamine and Atropine

McGlone R. et al., (1998) did not observe significant differences between intranasal midazolam and intramuscular ketamine and atropine during the suturing procedure in terms of the level of oxygen saturation, the behaviour of the child during recovery, adverse reactions and total time until discharge from the hospital, although, in this case, a somewhat lower median time was recorded in the group that received intranasal midazolam (75 min vs. 82 min) [7].

The combination of ketamine and atropine induced deeper sedation, which reduced the need for physical restraint of the children (14% vs. 86% of the midazolam group) (*p* < 0.01) [7]. This combination of drugs was preferred by parents and health personnel [7].

Sixty-six percent of children who received intranasal midazolam were resistant to receiving nasal drops [7].

### 3.5. Intranasal Midazolam versus Ketamine and Intravenous Midazolam

The initiation of sedation was faster in the group that received intravenous ketamine and midazolam, at 5.3 min on average (*p* < 0.001; 95% CI (3.2–7.4 min)), compared to the intranasal midazolam group [22]. The intravenous route was associated with deeper sedation, although, 35 min after administration, the levels of sedation became very similar in both groups [22].

The intranasal midazolam group was discharged from the hospital on average 19 min earlier than the group that received intravenous drugs (*p* < 0.02; 95% CI (4–33 min)) [22].

Parents and health professionals were more satisfied with sedation via the intravenous route [22].

Two-thirds of children who received intravenous drugs experienced random movements of the extremities [22]. No significant differences were observed in vital signs or adverse events.

### 3.6. Intranasal Midazolam versus Oral Diazepam

According to Everitt I. et al., (2002), the oral route was better tolerated than the intranasal route (*p* = 0.034) [8]. Oral diazepam was associated with a longer sedation time (31.0 ± 9 min) (*p* = 0.011) than intranasal midazolam (26.1 ± 9 min) [8], a longer time until hospital discharge and lighter sedation, according to the visual analogue scale administered by physicians, nurses, parents and researchers [8]. No adverse effects were recorded for any of the groups.

### 3.7. Intranasal Midazolam versus Intranasal Dexmedetomidine

Neville D. et al., (2016) compared two groups of children (one group received midazolam, and the other group received dexmedetomidine) and obtained the following results [10]: (a) positioning the child to perform the suture: 70% of children in the dexmedetomidine group showed no anxiety, compared to 11% of children in the midazolam group; (b) washing of the wound: 35% of children in the dexmedetomidine group showed no anxiety, compared to 6% of children in the midazolam group (OR = 9, 95% CI (1–84)).

Dexmedetomidine and midazolam behaved similarly, with no significant differences between the groups of children for all other measures recorded: anxiety at other times during the procedure, anxiety perceived by parents, satisfaction of parents and doctors, success of the suturing procedure, complications/adverse effects and total time to discharge [10].

### 3.8. Comparison among Three Different Volumes of Intranasal Midazolam

Tsze D. et al., (2017) observed slight differences (Table 6) in the initiation of sedation for the different volumes administered. All children received the same dose of 0.5 mg/kg midazolam [16]. No differences were observed among the three groups of children in terms of the level of distress, time to the start of the procedure, sedation level reached or adverse reactions [16].

### 3.9. Method of Administration of Intranasal Midazolam

In clinical trials published before 2002, intranasal midazolam was administered via the instillation of drops in the nostrils [5,7,8,19,20]; the exception was the study by Acworth JP. et al., in which, a spray device was used [22]. Subsequently published clinical trials used atomisation devices for intranasal mucosal administration [10,16,21].

### 3.10. Initiation of Action, Duration of Effect and Adverse Reactions

In the analysed studies, there were no significant differences in the time elapsed from the administration of the drug to the start of sedation and suturing among the intranasal route (intranasal midazolam), the intravenous route (ketamine + midazolam) and the intramuscular route (ketamine + atropine) [7,22].

For different volumes of intranasal midazolam, the time until the onset of the sedative effect was very similar [16].

Intranasal dexmedetomidine produced a deeper sedative effect during the initial stages of the suture procedure [10].

The combination of intravenous ketamine and midazolam and intramuscular ketamine and atropine showed deeper and more prolonged sedation [7,22].

No significant differences were found in the level of oxygen saturation, heart rate, respiratory rate or blood pressure between children who received intranasal midazolam and those who received the midazolam via other routes of administration or those who received other sedative drugs [7,8,10,20,21].

Two-thirds of the children who received intravenous ketamine and midazolam exhibited random movements of their extremities [22].

Table 7 shows the dose of intranasal midazolam, efficacy of sedation and number or children with nasal burning or irritation.

### 3.11. Satisfaction of Parents and Professionals

Three of the analysed studies did not assess parents’ and/or professionals’ satisfaction with the outcomes of sedation [5,8,20].

Theroux M. et al., (1993) observed a high degree of parental satisfaction with the use of intranasal midazolam compared with no drug or placebo [19].

Several studies showed that parents and health professionals prefer deeper sedation [7,22]. McGlone R. et al., (1998) observed that the preferred drug was the combination of ketamine and intramuscular atropine [7], while Acworth JP. et al., (2001) reported a preference for intravenous ketamine and midazolam [22].

Klein E. et al., (2011) found that parents would request sedation with intranasal midazolam in the future, indicating that they were satisfied with the sedation outcomes [21].

Neville D. et al., (2016) did not observe differences in the satisfaction of parents and health professionals with sedation by dexmedetomidine and midazolam [10].

## 4. Discussion

According to the analysed data, the intranasal administration of 0.2–0.5 mg/kg midazolam in healthy children at a paediatric emergency department achieved minimal/moderate sedation during the suturing procedure for an uncomplicated traumatic laceration.

### 4.1. Parenteral Route

Although intravenous sedation with midazolam is widely used in paediatrics [2] due to its rapid initiation of action, short duration and haemodynamic stability [23], the nasal mucosa is highly vascularised and provides direct access to the central nervous system, thereby avoiding the metabolism of the first-pass effect [9,11,16,24]. This causes a rapid initiation of action similar to that of the intravenous route [2], and a short duration of the effect (30–60 min) [2,11].

In the analysed clinical trials, the time passed from the administration of intranasal midazolam until the children were sedated (conscious sedation) was variable, lasting between 4.3 and 28 min [5,7,10,16,20,21,22]. However, in other studies, this time frame until the initiation of sedation ranged from 5–16 min, and adequate sedation was achieved in 7–10 min [11]. This rapid sedative action was also evidenced in the study by Mellion S. et al., (2017), which reported that, when 0.4 mg/kg intranasal midazolam was administered to children using an atomisation device, the drug reached a maximum concentration in blood in 10.1 min (interquartile range 9.7–10.8 min) and maintained a maximum plasma concentration above 90% from 5 to 17 min [9].

### 4.2. Oral and Intraoral Route

Regarding the oral and intraoral routes, in the analysed studies, there were no significant differences among the oral, intraoral (oral mucosa) and intranasal administration of midazolam in terms of the degree of anxiety reduction [8,20,21], the total length of hospital stay [8,20] and the occurrence of adverse effects. This was evidenced in other studies that considered these routes safe and effective for sedation in children [25]. However, several studies showed that the intranasal route presents a faster initiation of action and a better sedation outcome [21,25,26] and better recovery of the child [25,27].

The oral administration of midazolam requires the child’s cooperation in ingesting the drug and requires a higher dose than the intranasal route to achieve the same level of sedation [19,20,26,27]. This is due to the first-pass effect, which generates a bioavailability of 36–40% for oral midazolam, compared to 50–83% for intranasal midazolam [4,27,28,29]. Studies showed that satisfactory sedation was obtained in 100% of children treated with oral midazolam doses of 1 mg/kg [30]. However, other studies observed that, in order to achieve this rate of effectiveness, intranasal midazolam doses of 0.4–0.5 mg/kg were necessary [6]. Therefore, intranasal midazolam requires lower doses.

On the other hand, according to several of the analysed studies, the intranasal route is least tolerated by children due to local discomfort (burning or nasal irritation) [5,8,20,21,22] caused by the acidic pH of the drug (pH = 3.5) [19]. Especially in young children, this may suggest a need to temporarily restrain the child during the nasal administration of the medication [19].

Nasal discomfort was also evidenced in other studies [6,11,31]; however, a recent study showed that the nasal route was better tolerated than the oral route (due to midazolam’s bitter taste) [25].

The nasal burning produced by midazolam is mild and lasts only a few seconds [1], but is continuous if the drug is administered via nasal drops. When drops are used, more time is required for administration, and a portion of the dose may drain into the oropharynx and be absorbed enterally [6,32].

Nasal mucosa spray devices [33] convert the drug into an aerosol (with particles between 30 and 100 microns in size) [2]. Such devices have been shown to be effective for increasing the absorption of the drug, reducing loss through the oropharynx [9,21] and improving tolerance, since they decrease or even eliminate nasal discomfort [14,30]. According to several studies, the nasal irritation or burning caused by midazolam can be avoided by first administering aerosolised lidocaine [7,12,25,34]; attempts to alkalise the pH of the drug have not been successful [7].

To determine the necessary dose of a nasal spray, the 0.1 mL dead space of the device must be taken into account [10,14]. According to various studies, the ideal intranasal volume is 0.2 to 0.5 mL per nostril, with a maximum recommended volume of 1 mL per nostril [2,13,14,32].

The presence of nasal secretions caused by respiratory infections can hinder the absorption of the drug. This problem can be solved with a nasal wash with saline solution prior to the administration of the drug [2]. If the laceration is located in the child’s nose, the oral or intraoral route would be a more desirable option for achieving sedation [2,21].

### 4.3. Rectal Route

The rectal administration of midazolam presents an initiation of action, duration of effect and recovery time similar to the intranasal route, although a moderate effectiveness of 60–75% has been shown [35]. In addition, the child is not usually cooperative with rectal administration [7]. For these reasons, the rectal administration of midazolam is less common in paediatric emergency services [35].

### 4.4. Other Sedative Drugs

According to the analysed results, oral diazepam has a longer duration of sedation and lighter sedation [8]. Therefore, midazolam is preferable, since it is three to four times more potent and produces a higher level of anterograde amnesia [2] that prevents the patient from remembering the suture procedure as being unpleasant.

According to the analysed studies, intranasal dexmedetomidine has a better initiation of sedation during the initial phases of the suture procedure (positioning of the child, washing the wound) than intranasal midazolam; however, no significant differences were found for the rest of the procedure, recovery or the time elapsed until discharge from the hospital [10]. However, other clinical trials have shown significant differences in favour of intranasal dexmedetomidine in children (dose 1 µg/kg), which produces a greater reduction in anxiety and a better level of sedation than intranasal midazolam (dose 0.2 mg/kg) without adverse effects or nasal irritation or burning [36,37,38]; therefore, it may be preferable to midazolam. However, other studies have shown that the time to the initiation of sedation was longer with intranasal dexmedetomidine than with intranasal midazolam [13,38,39,40] and it produced longer-lasting sedation [13]. Therefore, new RCTs are needed to compare the effectiveness of different doses of dexmedetomidine and midazolam delivered to children using nasal mucosa spray devices.

Ketamine, which was used intravenously and intramuscularly in two of the studies included in this review [7,22], produced more profound and prolonged sedation than intranasal midazolam, although it presented a higher frequency of adverse effects. These results coincide with those of other studies that reported that intramuscular ketamine produced a deeper level of sedation and a longer recovery period [41] and more adverse effects [12,42]. Ketamine is not frequently administered intranasally because it leaves a bad taste in the mouth, even when it is administered with an atomiser [29]; additionally, high doses are required to produce sedative effects, and intranasal administration has a longer time until the initiation of action than intravenous administration [13].

### 4.5. Recommended Dose

In the analysed clinical trials, doses of intranasal midazolam between 0.2–0.5 mg/kg were considered effective. However, the intensity of sedation depends on the administered dose. Yelay D. et al. [6] analysed children with a mean age of 32 ± 9 months (age range 12 months to 6 years) and a weight of 14.5 ± 3 kg. One hundred percent of the children presented adequate sedation if they received 0.4–0.5 mg/kg intranasal midazolam, whereas, at doses of 0.2–0.29 mg/kg, the children were actively crying, struggling or requiring physical restraint [6]. Similar results have been shown in other studies that reported that 0.2–0.29 mg/kg of intranasal midazolam produced adequate sedation in 27% of children, whereas doses of 0.4–0.5 mg/kg produced adequate sedation in 100% of children [11].

In Spain, midazolam is marketed as an injectable solution, a buccal solution and oral tablets [43]. Currently, it is not marketed for rectal, intranasal or syrup administration. Therefore, the 5 mg/mL injectable solution or its equivalent should be used for intranasal administration. Due to the fact that this is a relatively low concentration, depending on the weight of the child, relatively high volumes may be required for intranasal administration.

Table A1 shows the pharmaceutical forms and concentration of midazolam currently sold in Spain [43].

### 4.6. Implications for Clinical Practice

Clinical nurses and physicians often perform the suturing of traumatic lacerations in the emergency department. For this procedure, health professionals have different drugs for sedation or analgesia, it being necessary to know the route for their administration. The rapid administration of drugs in the paediatric population in emergency situations is sometimes complicated for many reasons: agitation, problems with the insertion of intravenous catheters, impossibility of communication, etc. To avoid increasing the child’s discomfort by performing invasive techniques, we can use non-invasive routes, such as the intranasal route, to administer analgesic and/or sedative drugs to reduce the state of agitation or anxiety of paediatric patients, if this route is not contraindicated (epistaxis, rhinitis, nasal, facial or thoracoabdominal trauma, nasal obstruction, septal disorders, diseases affecting ciliary function such as cystic fibrosis, prior administration of vasoconstrictors and relative hypovolaemia) [44].

Based on our results, the intranasal administration of midazolam may be considered a good choice for children requiring suturing for traumatic lacerations due to its rapid sedative effect, similar to the effect obtained using the parenteral route and without significant adverse effects. However, this technique is easier to perform and has fewer complications. It also avoids traumatic procedures for children, such as needle sticks. It is true that, in older and heavier children, the amount of volume to be administered intranasally is greater, so this fact should be taken into account. 

### 4.7. Limitations of the Study

Regarding the limitations of the present study, it is worth noting the variability of the methods and scales used to assess the anxiety and sedation levels of the children in the clinical trials included in this review. Another limitation is the variability in the method of intranasal midazolam administration (drop instillation or administration with a spray device). In addition, not all studies included the location and size of the traumatic laceration that had to be sutured. Another limitation was the inclusion of low-quality articles due to the fact that one of the nine articles included showed a score < 3 on the Jadad scale. Additionally, it was not possible to verify the efficacy and safety of intranasal midazolam in comparison with other sedative drugs that are used in paediatrics (fentanyl, phenobarbital, thiopental, propofol, etc.) Given the absence of recent RCTs comparing the use of these sedatives in suture procedures, it would be wise to conduct more clinical trials in this area.

## 5. Conclusions

According to the analysed results and the doses of drugs used in the clinical trials included in this systematic review, we consider that the administration of intranasal midazolam in healthy children is a non-invasive method that is simple and effective (depending on the dose), producing a rapid state of minimal/moderate sedation in the child as part of a paediatric sedation and analgesic procedure. This sedative effect is similar to that produced using the intravenous route, which allows for the successful suturing of traumatic lacerations that do not present complications. Likewise, according to the results and the doses analysed, the intranasal midazolam is safe since it does not produce a significant adverse effect or require a lengthy recovery. 

In the analysed studies, children who received intranasal midazolam recovered rapidly, and the time until hospital discharge was similar to or even shorter than that of other drugs. However, this administration route sometimes requires the child to be restrained during the administration procedure. Health professionals’ prior training for its correct application is also necessary. In general, the intranasal route is a good option for short procedures such as suturing lacerations.

Additional good-quality RCTs are needed to compare intranasal midazolam with other sedative drugs used in paediatrics and to clarify the effectiveness and safety of midazolam versus intranasal dexmedetomidine at different doses in children.

## Figures and Tables

**Figure 1 children-09-00644-f001:**
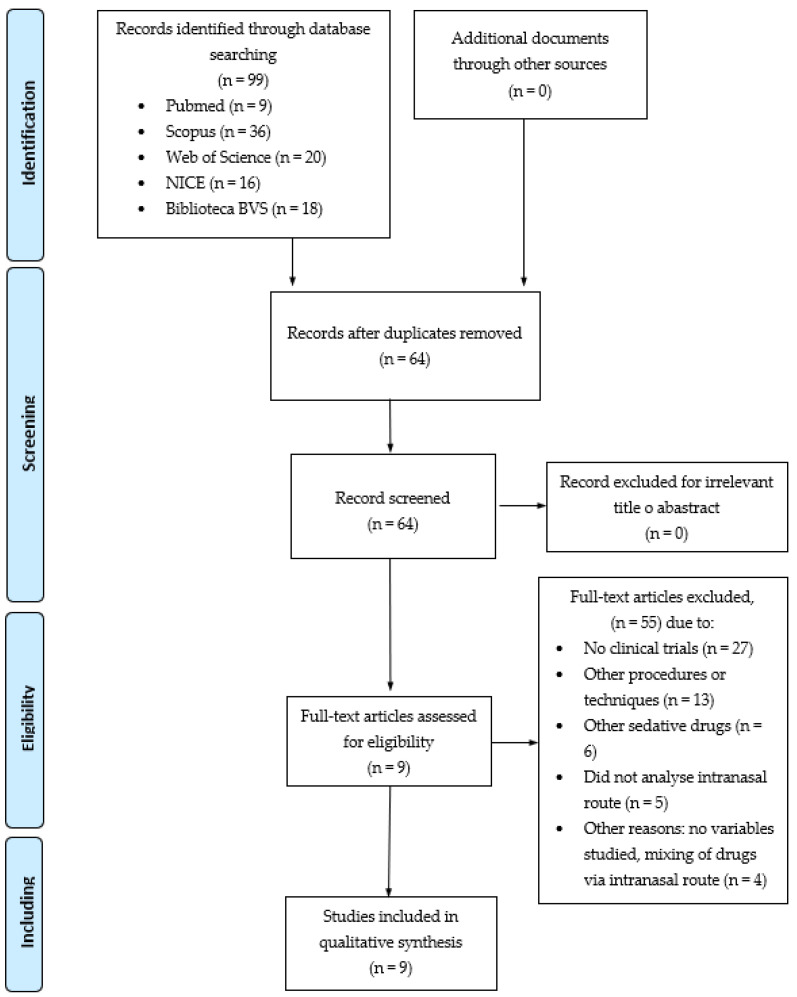
PRISMA flow chart (Moher et al., 2009) [17].

**Table 1 children-09-00644-t001:** PICO question structure.

P (Population)	(I) Intervention	(C) Comparison	(O) Outcome
Paediatric population.	Intranasal midazolam in sutures for traumatic lacerations.	Midazolam by other routes of administration, use of placebo and other sedative drugs.	To determine the efficacy and safety of intranasal midazolam in suturing procedures for traumatic lacerations in children.

**Table 2 children-09-00644-t002:** Inclusion/exclusion criteria.

Inclusion Criteria	Exclusion Criteria
Clinical trials	Studies conducted in animals
Studies on the administration of intranasal midazolam in traumatic laceration sutures	Studies that did not use intranasal midazolam
Population between 0 and <18 years	
English or Spanish language studies	

**Table 3 children-09-00644-t003:** Comparison of intranasal midazolam.

**Intranasal Midazolam**	**Versus**	Placebo (saline) and control group (no intervention) Haga clic o pulse aquí para escribir texto [19].
		Oral midazolam [8,20,21] and intraoral (oral mucosa) midazolam [21].
		Combination of ketamine and atropine intramuscular [7].
		Combination of ketamine and midazolam intravenous [22]
		Oral diazepam [8].
		Intranasal dexmedetomidine [10].
		Same dose of midazolam administered every 10–15 min [5].
		Comparison between three different doses of intranasal midazolam [16].

**Table 4 children-09-00644-t004:** Risk of bias and quality of the studies based on Jadad Scale.

Clinical Tial	Randomised Trial	Adequate Randomisation	Double Blind	Adequate Blinding	Abandonments and Withdrawals	Total Score
Theroux M.C. et al., (1993) [19]	1	1	1	1	1	5
Connors, K. et al., (1994) [20]	1	1	1	1	1	5
McGlone RG. et al., (1998) [7]	1	1	0	0	1	3
Lloyd CJ. et al., (2000) [5]	0	0	0	0	0	0
Acworth JP. et al., (2001) [22]	1	1	0	1	1	4
Everitt I. et al., (2002) [8]	1	1	0	1	1	4
Klein EJ. et al., (2011) [21]	1	1	0	1	1	4
Neville D. et al., (2016) [10]	1	1	1	1	1	5
Tsze D. et al., (2017) [16]	1	1	0	1	1	4

Score: 5 = excellent quality, 4 = good quality, 3 = acceptable quality, <3 = low quality.

**Table 5 children-09-00644-t005:** Characteristics of the included studies.

Clinical Trials	Sample	Size of Laceration	Dosage	Intranasal Midazolam	Sedation Onset Time	Time to Discharge from Hospital	Results and Conclusions	Adverse Events
Theroux M.C et al., (1993) [19]	n = 59; Children < 5 years	NA	3 groups: Group 27 children intranasal midazolam: 0.4 mg/ kg Group 17 children placebo (saline) Group 15 children control (no intervention)	Nasal drops	NA	NA	The group that received intranasal midazolam presented less crying (*p* < 0.003), less fighting (*p* < 0.04) and less need for physical immobilisation compared to the placebo and control groups.	2 children with unstable gait
Connors, K. et al., (1994) [20]	n = 58; Children aged 1 to 10 years	0.5 to 6 cm	2 groups: Group 27 children oral midazolam 0.5 mg/kg Group 27 children intranasal midazolam 0.25 mg/kg	Nasal drops	10 min in both groups	54 ± 15 min oral route 57 ± 16 min intranasal route	Both groups presented similar anxiety reduction (*p* < 0.05), without significant differences. The oral route and the intranasal route showed similar efficacy and safety in reducing anxiety in children. The intranasal route was more difficult to administer.	4 children with nasal burning (intranasal group)
McGlone RG. et al., (1998) [7]	n = 102 Children 1 to 7 years old	NA	2 groups: Group 50 children intranasal midazolam 0.5 mg/kg Group 50 children intramuscular ketamine 2.5 mg/kg + atropine 0.01 mg/kg	Nasal drops	10 min in midazolam intranasal	Mean 75 min midazolam group Mean 82 min ketamine group	Intranasal midazolam produced an effective sedation and amnesic effect (the children did not remember the suture). Ketamine produced dissociative anesthesia in most cases. No significant differences in SatO2, child recovery, time to discharge and observed adverse reactions were found.	Vomiting: 9 children in the ketamine group and 4 children in the midazolam group. Tearing: 6 children ketamine group and 13 children midazolam group. Increased salivation: 15 children ketamine group and 6 children midazolam group. Skin rash: 5 children ketamine group and 1 child midazolam group. Nightmares: 3 children ketamine group and 3 children midazolam group; and Unsteady gait in both groups with no significant differences.
Lloyd CJ. et al., (2000) [5]	n = 29 Children from 1.5 to 9.5 years of age	NA	1 group 29 children Initial dose midazolam 0.2 mg/kg if, after 10–15 min, there was no sedation response, the same dose was repeated up to a maximum of 0.5 mg/kg.	Nasal drops	Mean of 14 min	4 h	Intranasal midazolam doses of 0.2–0.5 mg/kg achieved adequate sedation in 22 children (76%) With this dose, sedation with intranasal midazolam is effective, safe and cost-effective.	38% children nasal burning. No other complications
Acworth JP. et al., (2001) [22]	n = 53 Children 0.5 to 12 years	NA	2 groups: Group 26 children intranasal midazolam 0.4 mg/kg Group 26 children 1 mg/kg intravenous ketamine + 0.1 mg/kg intravenous midazolam	Nasal spray	Midazolam group: mean 7.3 min Ketamine + midazolam group: mean 2 min (*p* < 0.001)	Midazolam group: mean 79 min Ketamine + midazolam group: mean 97.9 min (*p* = 0.02)	Adequate sedation to perform the suture in all of the children who received the combination of intravenous drugs (ketamine + midazolam) and in 24 of the 26 children who received midazolam was observed. The group of children who received the combination of intravenous drugs (ketamine + midazolam) presented deeper sedation, longer hospital stay and greater parents’ and staff satisfaction with the result of sedation.	Random movements in 17 children in the ketamine group. No significant differences in other adverse events
Everitt I. et al., (2002) [8]	n = 129 Children 1 to 5 years old	NA	3 groups: Group 42 children oral diazepam 0.5 mg/kg Group 45 children oral midazolam 1 mg/kg Group 42 children intranasal midazolam 0.4 mg/kg	Nasal drops	NA onset of action. Total duration of sedation: Oral diazepam: 31 ± 9 min Midazolam intranasal: 26.1 ± 9 min	Oral diazepam: 53.9 ± 16 min Intranasal midazolam: 48 ± 12 min	The oral route was better tolerated than the intranasal route. Sedation with midazolam (oral or intranasal) was more effective than with diazepam. Oral diazepam produced more prolonged sedation than intranasal midazolam.	2 children received midazolam (one oral and one intranasal) with crying episode after hospital discharge.
Klein EJ. et al., (2011) [21]	n = 177 Children 0.5 to 7 years	<4 cm: 165 children >4 cm: 4 children	3 groups: Group 59 children oral midazolam 0.5 mg/kg Group 59 children intraoral midazolam 0.3 mg/kg Group 59 children intranasal atomiser midazolam 0.3 mg/kg	Nasal atomiser	34 min Oral midazolam group 32 min Oral midazolam group 28 min Intranasal midazolam group	NA	The intranasal route showed a higher proportion of children with optimal sedation, a faster onset of action and a higher proportion of parents who would choose this route again. The intranasal route was the worst tolerated.	3 children, one in each group, had vomiting prior to discharge. Post-discharge nightmares in 1 child in the oral group and 1 child in the intranasal group.
Neville D. et al., (2016) [10]	n = 40 Children 1 to 5 years old	<5 cm	2 groups: Group 18 children midazolam intranasal 0.4 mg/kg Group 20 children dexmedetomidine intranasal 2 µg/kg	Nasal atomiser	NA A 30 min delay was allowed between administration and the start of the procedure.	Mean 2 h and 33 min dexmedetomidine group Mean 2 h and 24 min midazolam group	Intranasal dexmedetomidine and midazolam behaved similarly for anxiolysis in laceration suture procedures in children, except for children who received dexmedetomidine, who had less anxiety at the time of placement for the procedure.	2 children in group midazolam, one child had vomiting and one child had unsteadiness in ambulation.
Tsze D. et al., (2017) [16]	n = 99 Children 1 to 7 years old	<5 cm	3 groups: Same dose midazolam intranasal 0.5 mg/kg (starting at concentration 5 mg/mL) Group 33 children: volume 0.2 mL Group 33 children: volume 0.5 mL Group 33 children: volume 1 mL Depending on the weight of the child it was necessary to administer several doses. Maximum dose 10 mg (2 mL)	Nasal atomiser	Minimal sedation onset: Volume 0.2 mL: 4.7 min 95% CI (3.8–5.4 min) Volume 0.5 mL: 4.3 min 95%CI (3.9–4.9 min) Volume 1 mL: 5.2 min 95%CI (4.6–7 min)	NA	Similar clinical results for the three different doses of midazolam used were observed. Physicians were less satisfied with the result of sedation when the volume used was 0.2 mL.	3 children presented with vomiting and 2 children with inadequate sedation.

NA: Not available. According to the articles analysed, intranasal midazolam has the same effect as intranasal dexmedetomidine in healthy children, considering effectiveness and safety, allowing us to achieve minimal/moderate sedation, which facilitated the suture procedure of a traumatic laceration. Other drugs, such as oral diazepam, produced a more prolonged sedation. Ketamine produced a deeper, prolonged sedation and caused dissociative anesthesia.

**Table 6 children-09-00644-t006:** Starting sedation with different volumes of intranasal Midazolam.

Volume of Intranasal Midazolam	Start of Minimal Sedation *p* = 0.048 (Tsze et al., 2017) [16]
0.2 mL	4.7 min IC95% (3.8–5.4 min)
0.5 mL	4.3 min IC95% (3.9–4.9 min)
1 mL	5.2 min IC95% (4.6–7 min)

**Table 7 children-09-00644-t007:** Dose of intranasal midazolam, efficacy of sedation and adverse reaction (nasal burning).

Clinical Trials	Dose of Intranasal Midazolam	Efficacy of Sedation	Number of Children with Nasal Burning or Irritation
Theroux M.C et al., (1993) [19]	0.4 mg/kg	More than half of the children who received midazolam did not require physical immobilisation to perform the suture.	N.K
Connors, K. et al., (1994) [20]	0.25 mg/kg	Anxiety reduction was similar to the group receiving oral midazolam.	4 children of the 27 (14.8%)
McGlone RG. et al., (1998) [7]	0.5 mg/kg	Effective sedation and amnesic effect (the children did not remember the suture).	N.K
Lloyd CJ. et al., (2000) [5]	Initial 0.2 mg/kg; if, after 10–15 min, there was no adequate sedation, the dose was repeated up to a maximum of 0.5 mg/kg.	Adequate sedation in 22 of the 29 children (76%)	11 children of the 29 (38%)
Acworth JP. et al., (2001) [22]	0.4 mg/kg	Adequate sedation in 24 of the 26 children (92.3%)	N.K
Everitt I. et al., (2002) [8]	0.4 mg/kg	Adequate sedation in the 42 children who received intranasal midazolam.	N.K
Klein EJ. et al., (2011) [21]	0.3 mg/kg	Higher proportion of children with an optimal score on the activity scale (74%) compared to other routes of midazolam administration.	N.K
Neville D. et al., (2016) [10]	0.4 mg/kg	Similar sedative effect of intranasal midazolam in comparison with intranasal dexmedetomidine, except at the time of positioning the child to perform the suture.	N.K
Tsze D. et al., (2017) [16]	0.5 mg/kg	Similar sedation for the three volumes of intranasal midazolam administered (0.2/0.5/1 mL). Two children did not have adequate sedation	N.K

## Data Availability

Not applicable.

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
