# Peer review of "Evaluation of Intranasal Midazolam for Pediatric Sedation during the Suturing of Traumatic Lacerations: A Systematic Review"

_children, 2022, doi:10.3390/children9050644_

Round 1
Reviewer 1 Report
Thanks for the opportunity to review an interesting and relevant review article for the PEM world. However a few minor issues need to be addressed:
Introduction
Ll 36-41 seems somewhat out of proportion/ inadequate here. The paper looks at PSA for minor trauma repair (lip lacs) but not trauma in general. I suggest you give a succinct background (incidence, frequency in ED (!), associated waiting times if many unsuccessful PSA attempts etc., like line 62) for precisely lip lacs but not trauma in general.
Also ll 128-131 on page 16 in the discussion summarizes the problem and base for your study pretty well and should be described first thing in the introduction. (these lines are redundant to lines 239 ff in the discussion so would fit better in the intro).
LL 66-69 I suggest dividing this paragraph in two sentences.
Materials and methods
L 87: Please clarify here: Interventional clinical trials and did you include randomized trials only or even non randomized prospective ones (they would be interventional too though not randomized)? You state this in the abstract, just repeat this here for clarity.
L 115 Non clinical trial
L 117 there is a typo routel
L 128 How did you test the inter-reviewer reliability? Please describe
Table 3 there is a typo dexmedetomidine in English
Table 4 there seems to be lacking a term after “Abandonments and xyxy”
Table 5 please review and be consistent with the use of Majuscules and minuscules in medications and route of administration as well as with the use of full stops and commas (same for table 7) for numbers, there is a little bit of a mix in that table.
Acworth et al
what was the administration route for ketamine (in dosage)? I suppose IV as you mentioned this further up?
Please clarify ketamine + midaz vs midaz? (in results and conclusions)
Make “size of lacerations” column broader
Tsze
5 mgr/ml
Discussion
138-139 initiation of sedation of time sedation last? Reword to clarify please.
Ll 240-242 please divide sentence into 2 sentences and reword so that the meaning of the sentences are clear.
241 I suggest be consistent with using the word emergenca department (instead of ward)
L 248 “Epithiasis” do you mean epistaxis?
L 267 I believe “on the other hand” it should read “additionally”?
L 275 I think here it is decisive that you say rapid onset however minimal to morderate sedation when compared to IV (this is said in the next sentence)
L 285 I think even for short procedures midaz should be combined with LA instilled/injected into the wound? Please comment
Author Response
We appreciate very much your constructive comments, helpful information and your time. Thanks to this review, our manuscript was substantially improved. Responses to your comments are written in bold.
Reviewer 1
Thanks for the opportunity to review an interesting and relevant review article for the PEM world. However a few minor issues need to be addressed:
Thank you very much for your positive comments
Introduction
Ll 36-41 seems somewhat out of proportion/ inadequate here. The paper looks at PSA for minor trauma repair (lip lacs) but not trauma in general. I suggest you give a succinct background (incidence, frequency in ED (!), associated waiting times if many unsuccessful PSA attempts etc., like line 62) for precisely lip lacs but not trauma in general.
Thank you very much. We have clarified that these are minor traumas causing lacerations (line 42-43).
Also ll 128-131 on page 16 in the discussion summarizes the problem and base for your study pretty well and should be described first thing in the introduction. (these lines are redundant to lines 239 ff in the discussion so would fit better in the intro).
Thank you very much for your comment. The text on lines 128-131 of the discussion has been removed and moved to lines 56-59 of the introduction.
LL 66-69 I suggest dividing this paragraph in two sentences.
Thank you very much for your recommendations. However, this paragraph cannot be divided into two sentences because it briefly explains the objective of this study.
Materials and methods
L 87: Please clarify here: Interventional clinical trials and did you include randomized trials only or even non randomized prospective ones (they would be interventional too though not randomized)? You state this in the abstract, just repeat this here for clarity.
Thank you for your recommendation, "interventional randomized trials" has been included in this line.
L 115 Non clinical trial
Studies that were not clinical trials (randomised or not) were excluded.
L 117 there is a typo routel
Thank you, it was a mistake! It has been modified
L 128 How did you test the inter-reviewer reliability? Please describe
This sentence has been clarified. The disagreements and discrepancies were discussed by the reviewers until consensus was reached. If consensus was not achieved, another reviewer were consulted
Table 3 there is a typo dexmedetomidine in English
Thank you, the error has been corrected!
Table 4 there seems to be lacking a term after “Abandonments and xyxy”
Thank you, the error has been corrected!
Table 5 please review and be consistent with the use of Majuscules and minuscules in medications and route of administration as well as with the use of full stops and commas (same for table 7) for numbers, there is a little bit of a mix in that table.
Thank you, they have all been written in lower case and the numbers have been checked.
Acworth et al
thank you, it has been checked
what was the administration route for ketamine (in dosage)? I suppose IV as you mentioned this further up?
Thank you for your comment. The word intravenous has been added next to the Ketamine dose in Table 5 within the Acworth JP et al. clinical trial.
Please clarify ketamine + midaz vs midaz? (in results and conclusions)
Thank you very much for your comment. The results and conclusions corresponding to the combination of intravenous drugs (Ketamine 1mg/kg + Midazolam 0.1mg/kg) have been clarified.
Make “size of lacerations” column broader
Thank you for your comment. This column has been expanded.
Tsze
Thank you, the year of publication has been corrected.
5 mgr/ml
Thank you for your comment. The dose has been changed to 5 mg/ml.
Discussion
138-139 initiation of sedation of time sedation last? Reword to clarify please.
Thank you for your comment. The wording of these sentences has been modified to clarify the time elapsed from the administration of the drug (intranasal midazolam) until the children presented sedation (conscious sedation).
Ll 240-242 please divide sentence into 2 sentences and reword so that the meaning of the sentences are clear.
Thank you for your comment. The text has been divided into two sentences to improve the understanding of the text.
241 I suggest be consistent with using the word emergenca department (instead of ward)
Thank you for your comment. Text has been modified to include the words emergency department.
L 248 “Epithiasis” do you mean epistaxis?
Thank you for your comment. We have corrected the word.
L 267 I believe “on the other hand” it should read “additionally”?
Thank you for your comment. The text has been modified to include the word “additionally”.
L 275 I think here it is decisive that you say rapid onset however minimal to morderate sedation when compared to IV (this is said in the next sentence)
Thank you for your comment. Paragraph has been reworded to include that intranasal Midazolam produces minimal/moderate sedation (conscious sedation)
L 285 I think even for short procedures midaz should be combined with LA instilled/injected into the wound? Please comment
Thank you for your comment. This part of the text has been modified so as not to include conclusions that could confuse the reader.
Reviewer 2 Report
The comments had been addressed by the authors
Author Response
We appreciate very much your constructive comments, helpful information and your time. Thanks to this review, our manuscript was substantially improved. Responses to your comments are written in bold.
Reviewer 2
The topic is certainly relevant. It's not novel content, but I think particularly the findings related to dosing of intranasal midazolam would be beneficial to clinicians. Personally, for me, the biggest weakness of the paper is that the authors do not clearly define what they mean by "efficacy" and "safety". As you know, systematic reviews can have multiple outcomes or clinical questions that they attempt to answer, but it is important to have the outcomes clearly defined.
Thanks for your comments. Some sections of the study's conclusions have been modified to clarify the concept of “efficacy” (rapid sedative effect that produces conscious sedation, which makes it possible to successfully complete the suture of a traumatic laceration that does not present additional complications) and the concept of “safety” (no produce significant adverse effects and allow a rapid recovery of the child).
Secondly, they should clearly specify that the two reviewers were blinded to one another.
It has been included in the summary and in the study selection section.
Reviewer 3 Report
The authors analyze the practical problem of anesthesia in pediatric patients in the emergency room. Sewing wounds is the daily routine of emergency departments.
Analysis - a systematic review of nasal administration of midazolam in pediatric sedation will make it easier for emergency department doctors to reach for this preparation.
The authors correctly selected and analyzed the available literature from the PUBMED, SCOPUS WEB of Nauka NICE, VHL databases. They conducted this analysis in accordance with the Preferred Reporting Items.
It should be emphasized that two researchers analyzed independently and assessed the quality of the research using the Jadad scale.
Reliable analysis and clearly presented tables are the strong point of this publication.
A very good discussion supported by relevant references is a big plus of the job.
The authors also present the limitations of this analysis, pointing to the need for further clinical trials.
The presented conclusions show that midazolam administered intranasally in pediatric sedation is a good proposition allowing for quick achievement of the desired effect. Its administration by the nasal route is comparable to the parenteral route and is sufficient for suturing traumatic wounds with a low risk of adverse effects.
The work is a very good analysis that will facilitate the work of emergency department doctors
Author Response
We appreciate very much your constructive comments, helpful information and your time. Thanks to this review, our manuscript was substantially improved. Responses to your comments are written in bold.
Reviewer 3
The authors analyze the practical problem of anesthesia in pediatric patients in the emergency room. Sewing wounds is the daily routine of emergency departments.
Analysis - a systematic review of nasal administration of midazolam in pediatric sedation will make it easier for emergency department doctors to reach for this preparation.
The authors correctly selected and analyzed the available literature from the PUBMED, SCOPUS WEB of Nauka NICE, VHL databases. They conducted this analysis in accordance with the Preferred Reporting Items.
It should be emphasized that two researchers analyzed independently and assessed the quality of the research using the Jadad scale.
Reliable analysis and clearly presented tables are the strong point of this publication.
A very good discussion supported by relevant references is a big plus of the job.
The authors also present the limitations of this analysis, pointing to the need for further clinical trials.
The presented conclusions show that midazolam administered intranasally in pediatric sedation is a good proposition allowing for quick achievement of the desired effect. Its administration by the nasal route is comparable to the parenteral route and is sufficient for suturing traumatic wounds with a low risk of adverse effects.
The work is a very good analysis that will facilitate the work of emergency department doctors
Thank you very much for your positive comments.
This manuscript is a resubmission of an earlier submission. The following is a list of the peer review reports and author responses from that submission.
Round 1
Reviewer 1 Report
I would like to thank the authors for the effort to perform this systemic review, evaluating the efficacy and safety of intranasal midazolam. However, there are a few week points and queries, which need to be addressed:
Introduction: Very well written, nothing to complain
Material and Methods: Gives detailed information how the systemic review was performed, no missing information
Results: This is a very well performed study and the tables summarizing the results of the included papers give an excellent overview. However, in the end the chapter covering the results is too long. Following the tables there should be a short summery of the key findings of the evaluated studies and give the reader a clear message.
The total number of patients included is quite small. The studies compare a number of different drugs and modes of administration with intranasal midazolam or different doses of nasal midazolam with each other. This leaves the reader with the impression that the authors compare apples and oranges. On the other hand, a lot of information already given in the tables is repeated in the text modules of the results. Side effect such as burning or nasal irritation are mentioned, but should be summarized in a table. A table summarizing the used dosages and the achieved effectiveness (depth of sedation and reduction of anxiety) could be helpful.
Discussion: The discussion is quite long, unfortunately repeating again informations already given in the results. However, in the end it misses a clear summery of key points and recommendations for clinical practice. Furthermore there should be mentioned, that in older patients with higher weight the amount of volume becomes a problem. Since this paper does not realy add anything new it should critically discuss the indications and contraindications of this alternative route.
Conclusion: From clinical experience I have to disagree, that intranasal midazolam is simple to administer. Most patients have to be restrained (even if it is only a short moment), it is burning therefore it is always a problem if you need both nostrils to deliver the whole volume. Most children will not choose this method again. This needs to be mentioned. It is safe and effective, a good option in the outpatient department for very short procedures, such as suturing lacerations. The recommendation should be to combine it with an analgesic drug or infiltration of a local anesthetics.
Author Response
We appreciate very much your constructive comments, helpful information and your time. Thanks to this review, our manuscript was substantially improved. Responses to your comments are written in bold
Reviewer 1:
I would like to thank the authors for the effort to perform this systemic review, evaluating the efficacy and safety of intranasal midazolam. However, there are a few week points and queries, which need to be addressed:
Introduction: Very well written, nothing to complain
Thank you very much for your positive comments
Material and Methods: Gives detailed information how the systemic review was performed, no missing information
Thank you very much for your positive comments
Results: This is a very well performed study and the tables summarizing the results of the included papers give an excellent overview. However, in the end the chapter covering the results is too long. Following the tables there should be a short summery of the key findings of the evaluated studies and give the reader a clear message.
Thank you very much for your comment. At the end of Table 5, a brief summary of the most significant results has been included.
The total number of patients included is quite small. The studies compare a number of different drugs and modes of administration with intranasal midazolam or different doses of nasal midazolam with each other. This leaves the reader with the impression that the authors compare apples and oranges. On the other hand, a lot of information already given in the tables is repeated in the text modules of the results. Side effect such as burning or nasal irritation are mentioned, but should be summarized in a table. A table summarizing the used dosages and the achieved effectiveness (depth of sedation and reduction of anxiety) could be helpful.
Thank you very much for your comment. A new table has been included at the end of the results section (Table 7) summarizing the dose of intranasal Midazolam used in each study, the degree of efficacy of sedation and the percentage of children who presented nasal burning or irritation after Midazolam administration.
Discussion: The discussion is quite long, unfortunately repeating again informations already given in the results. However, in the end it misses a clear summery of key points and recommendations for clinical practice. Furthermore there should be mentioned, that in older patients with higher weight the amount of volume becomes a problem. Since this paper does not realy add anything new it should critically discuss the indications and contraindications of this alternative route.
Thank you very much! We have proceeded to summarize the information in Table 5 and the results section so that the data are not repeated now. We have added an "implications for clinical practice" section at the end of the discussion section where we have discussed indications and contraindications.
Conclusion: From clinical experience I have to disagree, that intranasal midazolam is simple to administer. Most patients have to be restrained (even if it is only a short moment), it is burning therefore it is always a problem if you need both nostrils to deliver the whole volume. Most children will not choose this method again. This needs to be mentioned. It is safe and effective, a good option in the outpatient department for very short procedures, such as suturing lacerations. The recommendation should be to combine it with an analgesic drug or infiltration of a local anesthetics.
Thank you very much for your comments. The authors refer to the fact that it is easy to apply compared to other invasive techniques such as venous catheter insertion. However, we have added your recommendations in the conclusions section.
Reviewer 2 Report
This is a nice description of using IN midazolam for laceration repair but offers nothing particularly new or novel in my opinion as it as you point out cannot address safety or even really quality systematically
Author Response
We appreciate very much your constructive comments, helpful information and your time. Thanks to this review, our manuscript was substantially improved. Responses to your comments are written in bold.
Reviewer 2
This is a nice description of using IN midazolam for laceration repair but offers nothing particularly new or novel in my opinion as it as you point out cannot address safety or even really quality systematically
Thank you very much for your comments. This is the first systematic review to look at this problem (registered in PROSPERO with number CR42021224635). Clinical nurses and physicians often perform suturing of traumatic lacerations in the emergency department and may use different drugs for sedation or analgesia and the route for their administration. Based on our results, intranasal administration of midazolam can be a good choice for children requiring suturing for traumatic lacerations due to appears to have a rapid sedative effect similar to the parenteral route and without significant adverse effects. However, this technique is easier to perform and with fewer complications avoiding traumatic procedures for children such as needle sticks.
Reviewer 3 Report
Dear Authors
I read with pleasure this systematic review. Except for risk bias, on which I can not comment, I think the study is written very well and gives a big amount of clinically valuable information
I have a couple of minor comments:
1. Table 3 includes only 8 but not 9 studies, please correct or explain.
2. Study by Lloyd et all. 2000 has a total score of quality 0, please discuss the inclusion of low quality articles (<3 score) in this review in the discussion or limitation section.
Author Response
We appreciate very much your constructive comments, helpful information and your time. Thanks to this review, our manuscript was substantially improved. Responses to your comments are written in bold.
Reviewer 3
Dear Authors
I read with pleasure this systematic review. Except for risk bias, on which I can not comment, I think the study is written very well and gives a big amount of clinically valuable information
I have a couple of minor comments:
1.Table 3 includes only 8 but not 9 studies, please correct or explain.
Thank you very much. It was a mistake, we have included the missing reference (nº 10).
- Study by Lloyd et all. 2000 has a total score of quality 0, please discuss the inclusion of low quality articles (<3 score) in this review in the discussion or limitation section.
Thank you. We have done it, adding a paragraph in the limitation section!
Round 2
Reviewer 2 Report
None